# Exploratory Assessment of Proteomic Network Changes in Cerebrospinal Fluid of Mild Cognitive Impairment Patients: A Pilot Study

**DOI:** 10.3390/biom13071094

**Published:** 2023-07-08

**Authors:** Aida Kamalian, Sara G. Ho, Megha Patel, Alexandria Lewis, Arnold Bakker, Marilyn Albert, Richard J. O’Brien, Abhay Moghekar, Michael W. Lutz

**Affiliations:** 1Department of Neurology, Johns Hopkins University School of Medicine, Baltimore, MD 21224, USA; akamali4@jhmi.edu (A.K.);; 2Department of Psychiatry and Behavioral Sciences, Johns Hopkins University School of Medicine, Baltimore, MD 21287, USA; 3Department of Neurology, Duke University School of Medicine, Durham, NC 27710, USA

**Keywords:** Alzheimer’s disease, mild cognitive impairment, proteomics, cerebrospinal fluid, neuroinflammation

## Abstract

(1) Background: Despite the existence of well-established, CSF-based biomarkers such as amyloid-β and phosphorylated-tau, the pathways involved in the pathophysiology of Alzheimer’s disease (AD) remain an active area of research. (2) Methods: We measured 3072 proteins in CSF samples of AD-biomarker positive mild cognitive impairment (MCI) participants (*n* = 38) and controls (*n* = 48), using the Explore panel of the Olink proximity extension assay (PEA). We performed group comparisons, association studies with diagnosis, age, and APOE ε4 status, overrepresentation analysis (ORA), and gene set enrichment analysis (GSEA) to determine differentially expressed proteins and dysregulated pathways. (3) Results: GSEA results demonstrated an enrichment of granulocyte-related and chemotactic pathways (core enrichment proteins: ITGB2, ITGAM, ICAM1, SELL, SELP, C5, IL1A). Moreover, some of the well-replicated, differentially expressed proteins in CSF included: ITGAM, ITGB2, C1QA, TREM2, GFAP, NEFL, MMP-10, and a novel tau-related marker, SCRN1. (4) Conclusion: Our results highlight the upregulation of neuroinflammatory pathways, especially chemotactic and granulocyte recruitment in CSF of early AD patients.

## 1. Background

Alzheimer’s Disease (AD) is a progressive neurodegenerative disease characterized by progressive cognitive decline and the distinctive presence of amyloid-β (Aβ) plaques and neurofibrillary tangles (NFTs) during histopathological examination [1]. Numerous studies have reported that levels of AD-relevant biomarkers such as phosphorylated-tau at threonine 181 (p-tau181) [2], Aβ_1–40_ [3], and Aβ_1–42_ [3] in cerebrospinal fluid (CSF) and plasma are associated with AD diagnosis. However, despite the well-established presence of Aβ plaques, tau-containing NFTs and neurodegeneration as hallmarks of AD, the upstream mechanisms leading to accumulation of these proteins that give rise to the disease are still unclear [4].

Mild cognitive impairment (MCI) refers to the transitional stage between healthy aging and dementia where the earliest symptoms of cognitive decline occur while daily functions are still preserved [5]. Protein profiles of Aβ-positive or amnestic MCI patients have been explored recently in order to identify the intricate biology underlying AD in its early stages before extensive neurodegeneration takes place [6].

To investigate upstream mechanisms of AD pathophysiology and the associated protein profile, systems-based approaches, such as proteomics, have been employed [7,8]. Among proteomics techniques, the Olink proximity extension assay (PEA) provides a highly accurate, high-throughput measurement of hundreds to thousands of proteins. Network proteomic analyses have collectively identified differentially expressed proteins involved in a wide range of intra- and extracellular pathways including synaptic function, inflammation, sugar metabolism, and astrocyte/microglial pathways in AD dementia and MCI [9,10,11,12]. Considering the lack of treatment options for AD and the complex pathophysiology of the disease, more studies with deeper proteome depth are needed on the network proteomics of MCI patients in order to facilitate biomarker discovery and the identification of drug targets in perturbed pathways.

In this pilot study, we used the Explore Olink PEA assay to quantify the differential expression of approximately 3000 proteins in CSF of AD biomarker-confirmed (low CSF Aβ_1–42_/Aβ_1–40_ and high CSF p-tau181) individuals with MCI compared to controls.

## 2. Materials and Methods

### 2.1. Participants

Participants included subjects enrolled in the Johns Hopkins Alzheimer’s Disease Research Center and the Center for CSF Disorders in the Dept of Neurology who were cognitively normal or met the criteria for MCI. The clinical diagnostic classification followed the recommendations of the National Institute on Aging/Alzheimer’s Association workgroups [13].

### 2.2. Consent Statement

All subjects gave their informed consent for inclusion before they participated in the study. This study was conducted in accordance with the Declaration of Helsinki, and the protocol was approved by institutional review board of Johns Hopkins University.

### 2.3. CSF Collection and CSF Biomarker Assays

CSF was collected from 48 cognitively normal and 38 MCI participants. Participants underwent a lumbar puncture and blood collection in the fasted state. Twenty ml of CSF was directly collected in a 50 mL polypropylene tube and transported on ice to the lab along with 20 mL of whole blood in EDTA tubes, where they underwent centrifugation at 2500× *g* for 15 min. Samples were then aliquoted in 0.5 mL aliquots and frozen at −80 C within 2 h of collection. CSF Aβ_1–42_, Aβ_1–40_, and p-tau181 were measured using the Lumipulse G1200 assay (Fujirebio, Malvern, PA, USA). The intra-assay coefficients of variation for this assay were 3.4% for Aβ_1–42_, 2.7% for Aβ_1–40_, and 1.8% for p-tau181. The ratio of CSF Aβ_1–42_/Aβ_1–40_ and p-tau181 were used in the current analyses. The participants with an Aβ_1–42_/Aβ_1–40_ ratio below 0.068 and p-tau181 levels above 50.6 pg/mL were identified as AD-biomarker positive.

All MCI participants selected for this study were required to be AD-biomarker positive based on their CSF values, whereas all controls were required to be AD-biomarker negative based on the cut-offs established by Greenberg et al. [14].

### 2.4. Protein Measures Using Olink Proximity Extension Assay (PEA), Quality Control (QC) and Data Pre-Processing

The Olink Explore panel, a highly sensitive and specific technique, measures the expression of 3072 proteins in 10 µL CSF (Olink, Uppsala, Sweden). CSF protein measurements were conducted using PEA technology, following the manufacturer’s protocol [15].

Pre-processing handling of data included plate-based normalization and QC checks based on appropriate Olink protocols [15]. Outlier deletion was performed subsequently by detecting and deleting datapoints that were above or below 5 SD of mean normalized protein expression (NPX) of each assay (protein). All datapoints with QC or assay warning were also deleted.

### 2.5. Group Comparisons

Statistical analyses were performed with R software (R-project.org) using the publicly available OlinkAnalyze package. Groups were compared using Welch’s two-sample, two-sided *t*-test analysis and Benjamini–Hochberg (BH) post hoc analysis. Differentially expressed proteins were defined as assays with false discovery rates (FDR) below 0.05 (FDR-adjusted *p*-value < 0.05).

### 2.6. Multivariate Association Analysis of CSF Protein Levels with Age, Sex, Diagnosis, and APOE Genotype

Multivariate linear regression models were used to test for the association of CSF mean NPX values of each protein with each covariate (age, sex, diagnosis, and APOE genotype) and for interactions between the covariates, specifically age with diagnosis (See Appendix A). All statistical analyses were completed using SAS (version 9.4, SAS Institute, Cary, NC, USA) and JMP (version 16.2.0, SAS Institute, Cary, NC, USA) software.

### 2.7. Overrepresentation Analysis (ORA)

After performing differential abundance analysis on our proteomics dataset, we conducted overrepresentation analysis (ORA) to identify differentially expressed pathways. This bioinformatics analysis helped us identify pathways that were enriched with significant changes in protein abundance. The goal of this analysis was to gain a better understanding of the functional implications of the differentially expressed proteins and to identify potential biological processes involved in the observed phenotypes.

We performed overrepresentation analysis on differentially expressed proteins (with differential abundance of FDR < 0.01) using Gene Ontology (GO) biological process and the Wikipathway database [16]. ORA was performed using the WEB-based GEne SeT AnaLysis Toolkit (WebGestalt) [17], which is a functional enrichment analysis web tool implementing several biological functional category databases such as KEGG, Reactome, WikiPathway, and PANTHER [17]. The default settings of WebGestalt were used with the FDR cut-off of 0.05.

### 2.8. Gene Set Enrichment Analysis (GSEA)

Gene Set Enrichment Analysis (GSEA) is a computational method that determines whether the expression of a set of genes is significantly different between two phenotypes [18]. GSEA was performed using GSEA 4.3.2 software [19,20]. The GSEA calculates the signal-to-noise ratio for all proteins and orders gene sets by normalized enrichment scores (NESs). We performed the GSEA with the default settings of the software, which included 1000 permutations, phenotype permutation type, exclusion of gene sets larger than 500 and smaller than 15, and using weighted enrichment statistics. Gene set databases (taken from MSigDB version 2022.1.Hs) used in this analysis included: KEGG, BioCarta, PID, Reactome, and Wikipathways.

In addition to ORA, we also performed GSEA in our proteomics study of CSF from MCI patients. We chose to use both methods because they provide complementary information and can help to validate each other’s results.

ORA allowed us to identify pathways that are overrepresented in our list of differentially expressed proteins, whereas GSEA enabled us to assess the enrichment of pre-defined gene sets in the list of differentially expressed proteins ranked by normalized enrichment scores, which can provide a more comprehensive understanding of the biological processes affected by the disease.

Using both methods also helped to overcome the limitations of each method, such as the dependence on pre-defined gene sets in ORA and the need for a large reference gene set in GSEA. Overall, by using both ORA and GSEA, we were able to identify novel and previously known pathways that are dysregulated in MCI and gain a better understanding of the underlying mechanisms contributing to the pathogenesis of this disease.

## 3. Results

### 3.1. CSF Proteomic Group Differences in AD-Biomarker Positive MCI Compared to HC

We used PEA technology with the Olink Explore panel to determine the differential expression of approximately 3072 proteins in AD-biomarker positive MCI patients compared to the healthy controls (Table 1). At least 96% of the datapoints in each of the Explore Olink panels passed the QC thresholds. The intra-assay and inter-assay coefficient of variation (CV) of the panels did not exceed 15%. Only one protein did not pass the Olink batch release quality control criteria and was excluded from the study (KNG1).

After the pre-processing steps, 2936 proteins were assayed in CSF. Among these, 117 proteins were found to be differentially expressed in CSF based on group comparisons (FDR-adjusted *p*-value < 0.05; Appendix A). Only three of these proteins were downregulated in MCI patients (Figure 1).

The results of the group comparison conducted between participants younger and older than the median age (71 years old), demonstrated that no proteins were differentially expressed in CSF of younger participants compared to the older participants (Figure 2, Appendix A).

Even though, the overrepresentation analyses of the differentially expressed proteins using GO databases did not show any pathways to be significantly enriched among these differentially expressed proteins, it is worth noting that “microtubule polymerization or depolymerization” and “leukocyte activation involved in inflammatory response” were among the top two biological processes that were found to be enriched in the CSF proteins in MCI. The proteins contributing to these terms included: ITGAM, ITGB2, TGFB2, MAPT, HSPA1A, MAP2, and SNCA. ORA performed using the Wikipathways database did not yield FDR-significant results; however, the “Microglia Pathogen Phagocytosis Pathway” was the top enriched biological process in CSF. The proteins that contributed to the enrichment of these pathways included: ITGB2, ITGAM, C1QA, TREM2, MAPT, CRKL, and MAP2K1.

### 3.2. Associations with Age, Sex, Diagnosis, and APOE Genotype

Multivariate linear regression analysis of individual proteins with covariates of age, diagnosis, sex, and APOE genotype revealed the assays that were associated with each of these variables and a full model including all covariates in CSF (Appendix A).

The multivariate analysis identified 147 proteins in CSF that were significantly associated with diagnosis (FDR-adjusted *p*-value < 0.05; Appendix A). The multivariate analysis identified 88 proteins that were significantly associated with age and 33 proteins that were associated with both diagnosis and age (Appendix A).

ORA of CSF proteins that were significantly associated with age using GO terms did not yield any FDR-significant results. However, “cell chemotaxis” (enrichment ratio [ER] = 2.6) was the most enriched term among age-associated proteins. Enrichment ratio is the ratio of the number of observed, differentially expressed genes of a certain pathway divided by the expected value if the pathway was neither enriched nor downregulated. Many chemokine motifs were among assays that contributed to this result. Moreover, C1QA, ITGAM, and ITGB2 are prominent complement-related biomarkers seen in this list.

The most enriched biological processes among diagnosis-associated assays were “inclusion body assembly” (ER = 7.7; including heat shock proteins such as: DNAJA4, DNAJB2, HSPA1A, and MAPT) and “apoptotic mitochondrial changes” (ER = 4.4). Nonetheless, they were not found to be FDR-significant.

Interestingly, the APOE genotype (i.e., at least one ε4 allele present) was significantly correlated with APOE and FGFBP1 in CSF (Appendix A).

### 3.3. Protein List Overlap between Group Comparison and Multivariate Association Analysis

A substantial overlap is observed between the group comparison results and association studies in terms of differentially expressed proteins. Of note, SCRN1, TREM2, and MMP-10 were significant in both Welch’s *t*-test and the association analysis of diagnosis in the multivariate association study, with and without interactions (Appendix A). Other proteins such as ITGAM, ITGB2, C1QA, GFAP, and NEFL, which were identified as differentially expressed in CSF of MCI participants in the group comparison, were also significantly associated with both diagnosis and age in the multivariate models (Appendix A).

### 3.4. Gene Set Enrichment Analysis Results

The only pathway found to be significant among all canonical pathway databases was the granulocytes pathway in BioCarta (FDR-adjusted *p*-value = 0.052; enrichment score (ES) = 0.75; Figure 3). The enrichment score (ES) is the maximum deviation from zero (random distribution) encountered during the GSEA analysis and reflects the degree to which the genes in a specific gene set are overrepresented at the top or bottom of the entire ranked list of genes. The core enrichment proteins that most significantly contributed to the significance of this pathway included: ITGB2, ITGAM, ICAM1, SELL, SELP, C5, and IL1A.

The list of protein name, gene name, and UniProt ID of all unique proteins measured by Olink Explore panel can be found on Appendix A.

## 4. Discussion

The most significant upregulated protein in CSF of the MCI patients compared to controls was secernin-1 (SCRN1), which was associated with diagnosis but not age. Secernin-1 is a novel and specific phosphorylated tau binding protein that has been recently shown to be abundantly present in amyloid plaques and NFTs [21]. Little is known about the function of secernin-1. However, a recent study that compared AD brain histopathology with other tauopathies demonstrated that secernin-1 is specific for tau isoforms in AD dementia and Down Syndrome as opposed to those in frontotemporal lobar degeneration dementia with Lewy bodies, progressive supranuclear palsy, and corticobasal degeneration [22]. Moreover, SCRN1 was found to have a strong, non-linear correlation with p-tau181 concentrations in CSF of our participants (correlation with log_2_ of p-tau181 = 0.8, *p*-value < 0.0001; Appendix A). Therefore, secernin-1 may act as a specific AD biomarker in CSF even at early stages of the disease. To the best of our knowledge, this study is the first to identify this differentially expressed tau binding protein in CSF.

This study also identified pathways that were enriched in MCI participants compared to controls (Figure 4). Most importantly, neuroinflammatory pathways (i.e., microglial pathogen activation and chemokine-related pathways) were the most enriched pathways based on ORA and GSEA. There has been considerable evidence pointing to the involvement of inflammatory processes during the early phases of AD, including the pruning of complement-marked synapses by reactive, disease-associated microglia activated by a cocktail of chemokines in AD pathogenesis [23,24]. This mechanism is activated by C1QA-coated synapses being identified by ITGAM/ITGB2 (CR3) heterodimer receptors on microglia that are activated by chemokines to phagocytize synapses [23,25,26]. TREM2 and AGER are also recognized as C1QA receptors that could induce phagocytosis, oxidative burst, and migration of inflammatory cells [23,25]. Considering that all these elements are significantly upregulated in CSF of MCI patients, our results support a role for this mechanism in AD pathogenesis. Deeper proteome depth thus could enable measurement of both of the key activators (complement and their respective ligands) to better identify the precise role and timing of the important hub proteins in neuroinflammation in different phases of neurodegeneration. Moreover, our GSEA results and ORA highlight an increased granulocyte activation and chemotactic activity in the CSF of patients, which has been indicated by the literature previously [27,28]. It is noteworthy that many of the above-mentioned markers are found to be associated with both age and diagnosis in our multivariate analysis. This observation might be explained by the inflammaging hypothesis, which recognizes an increased level of pro-inflammatory markers with age that increases the susceptibility to and is accentuated in dementia [29].

Dysregulation of the mitogen-activated protein kinase (MAPK) pathway has been demonstrated in AD for several decades [30,31]. MAPT (hyperphosphorylated tau) is the most abundant protein in NFTs, and MAPK signaling is suggested to affect β- and γ-secretase activity, cause neuronal apoptosis, and even induce pro-inflammatory cytokine upregulation in CNS [32,33]. One of components of the MAPK pathway, MAP2K1, induces p53 and cdk5 activation, which are two of the main tau kinases [34]. MAPT, MAP2, and MAP2K1 have been differentially expressed in the CSF of our patients. Thus, combined with secernin-1, this depth of proteome analysis facilitates the measurement of all features in a pivotal pathway starting from substrate (MAPT and MAP2), involving their specific kinases (MAP2K1) to the specific phosphorylated tau binding protein (secernin-1).

The third mechanism prominent in our results is the cellular chaperone network. Chaperones (e.g., HSPA1A, HSPB6, DNAJC6, DNAJA2, and DNAJA4) have been shown to be upregulated in AD animal models as a neuroprotective mechanism inhibiting the aggregation of tau and Aβ fibril formation [35,36,37]. It is also suggested that some of these chaperones (e.g., DNAJA1; not significant in our results) facilitate Aβ toxicity by stabilizing Aβ_1–42_ oligomers and inducing mitochondria-dependent cell death [36]. Considering that “inclusion body assembly” and “apoptotic mitochondrial changes” were also the top two diagnosis-associated ORA findings of our study, further studies are needed to validate the protective and/or detrimental effects of chaperone families in AD.

Oligodendrocyte myelin glycoprotein (OMG) is among the three proteins found to be downregulated in CSF in participants with MCI compared to controls. As evident from its name, OMG is suggested to have a role in regulation of collateral sprouting of intact axons in response to injury and is restricted to oligodendrocytes in the brain [38]. Zhang et al. report OMG as a hub protein involved with myelin sheaths that has a negative correlation with amyloid plaques and NFTs in the AD brain [38,39]. Furthermore, ARHGAP30 transcription was found to be related with mature microglia-specific homeostatic surveillance [40]. Although this protein is not well-characterized in AD, downregulation of this protein might indicate a loss of homeostatic surveillance on microglia in AD. However, the role of these downregulated proteins (ARHGAP30 and KIAA2013) in AD needs further investigation. This is the first account of the significant downregulation of these proteins in CSF of MCI patients.

Multiple studies have assessed the proteomics of CSF in AD dementia patients using Olink platforms [8,11,30,41,42]. However, there were only a few that looked at cases with mild AD or MCI [8,11,42]. These studies were limited to a specific subset of Olink panels, which quantified approximately 90 and 300 assays, respectively. The status of AD biomarkers in the patients, the participants’ inclusion criteria, and downstream statistical analysis were also different in each study. With these considerations in mind, several pathways and specific proteins continued to come up in these papers (e.g., MMP-10, SMOC2, EZR).

Apart from other Olink studies, many CSF proteins determined through our experiments have been previously reported in other proteomics studies of AD (e.g., NEFL, MMP-10, TREM2, EIF4EBP1, GFAP, SMOC1, SMOC2, MAPT, FABP3, CHI3L1, EZR) [8,11,26,42]. We also identified members of heat shock proteins, previously recognized in NFTs, to be upregulated in CSF of patients (HSPA1A) [21]. Therefore, because of the proteome depth of our experiments, we were able to recapitulate previously identified protein profiles of AD pathogenesis as well as novel proteins, which could be putative biomarkers with further validation studies.

A major limitation of our study is that the control and MCI groups differed by age. While we did not see any differential expression of proteins in CSF on the basis of age alone in the group comparison performed with Welch’s *t*-test, we cannot exclude the effect of age on our differential analysis by diagnosis. Multivariate analysis, including age as a covariate, identified proteins that showed a statistically significant effect with age for this sample. Moreover, we acknowledge the limitation of the relatively small sample size of our study. We conducted a power calculation for our study, which can be found in the Appendix A, to ensure that we had sufficient statistical power to detect meaningful differences. It is worth noting that similar sample sizes have been used in previous studies in the field, and this is a relatively common limitation in proteomic studies of CSF [30,41]. Nonetheless, future studies with larger sample sizes are needed to validate our findings and provide more comprehensive insights into the molecular mechanisms underlying cognitive impairment. An additional limitation of every proteomics study is the inability of current high-throughput methods to measure every protein present in the proteome. Therefore, our ability to map the proteomics network of disease is restricted to the scope of the current methods. Moreover, it is crucial to note that downstream pathway analysis methods such as ORA and GSEA investigate the enriched protein list against the known, existing pathways in their databases. Therefore, novel enriched pathways not in the current databases are potentially missed by these methods [18]. In addition, the lack of gold standard datasets leads to heterogenous pathway analysis results when using different existing databases [18]. Thus, our findings should be viewed as a hypothesis-generating, pilot study with a limited number of participants that needs to be validated with a larger cohort to increase the power of the study.

## 5. Conclusion

In conclusion, we identified both well-established and novel markers, such as the tau-binding protein secernin-1, in CSF of MCI participants compared to controls. The utilization of pathway analysis methods further highlighted the role of neuroinflammation and especially opsonization of complement-marked synapses by macrophages in early AD biology. Other pathways, including protein misfolding and microtubule assembly pathways, were also implicated based on our results. Taken together, these findings emphasize the involvement of several pathways in early AD pathogenesis and protein markers associated with it, which require further validation.

## Figures and Tables

**Figure 1 biomolecules-13-01094-f001:**
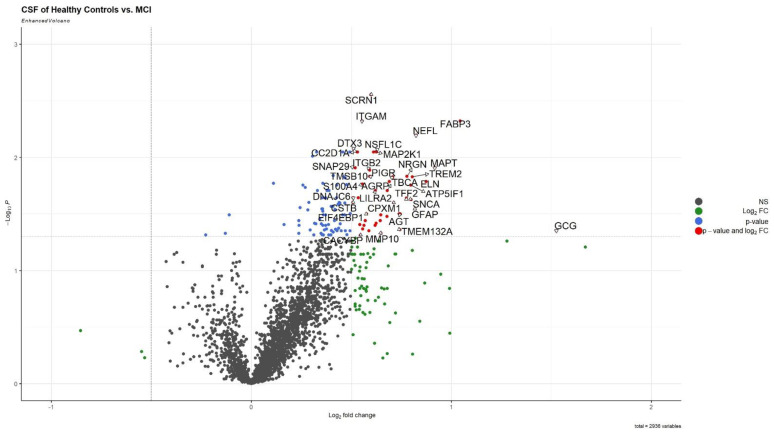
Volcano plots of differentially expressed proteins in cerebrospinal fluid of MCI compared with HC (MCI: mild cognitive impairment, HC: healthy control). The triangles in the figure connect the label of the proteins with their location on the volcano plot.

**Figure 2 biomolecules-13-01094-f002:**
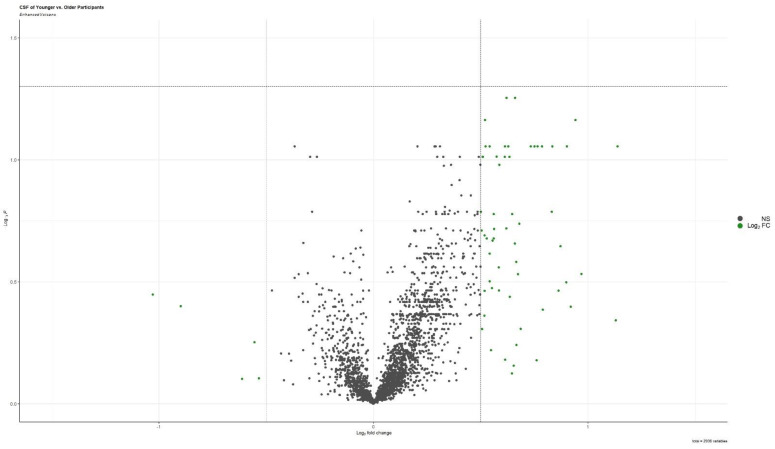
Volcano plots of differentially expressed proteins in cerebrospinal fluid of participants younger than the median age compared with older participants.

**Figure 3 biomolecules-13-01094-f003:**
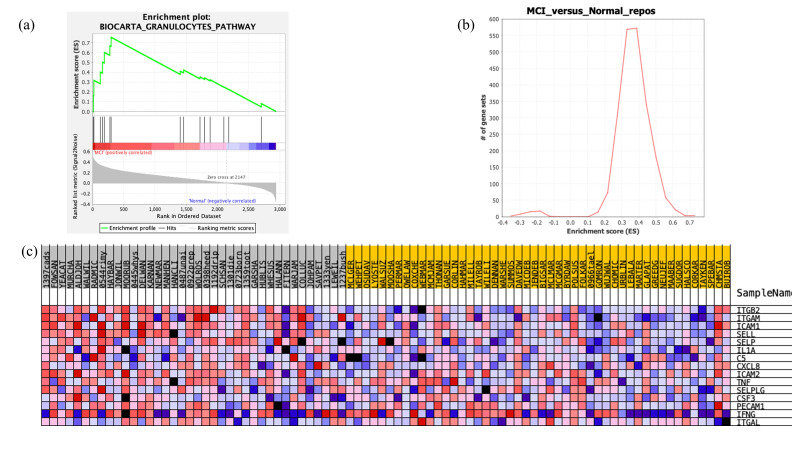
Results of the gene set enrichment analysis conducted with the GSEA software on CSF expression data. (**a**) Enrichment plot, (**b**) enrichment score map with the enrichment score as *x*-axis and number of gene sets as the *y*-axis, and (**c**) heatmap of the granulocyte pathway genes in BioCarta database (FDR-adjusted *p*-value = 0.052; enrichment score = 0.75). The subject IDs labeled in grey demonstrate mild cognitive impairment, and the yellow labels mark the healthy control subject IDs.

**Figure 4 biomolecules-13-01094-f004:**
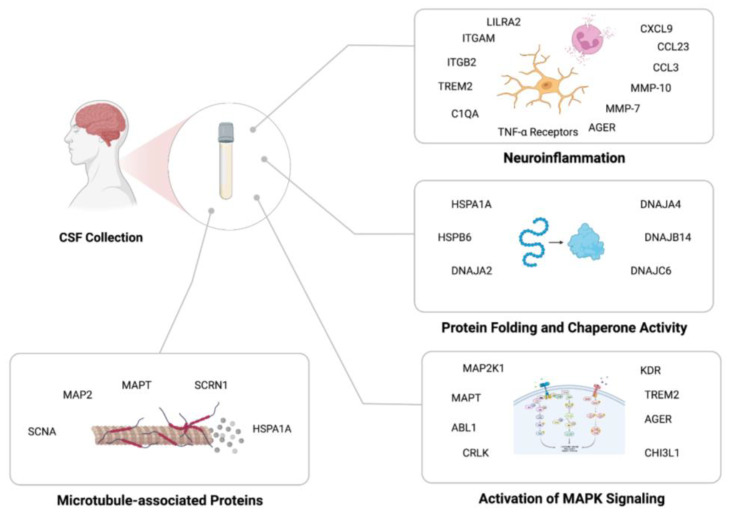
An overview of the enriched proteins and the related processes in cerebrospinal fluid of mild cognitive impairment patients compared to controls (Created with BioRender.com, accessed on 6 February 2023).

**Table 1 biomolecules-13-01094-t001:** Demographic characteristics of study participants and the concentrations of cerebrospinal fluid biomarkers in mild cognitive impairment and healthy controls (HC: healthy control, MCI: mild cognitive impairment, SD: standard deviation, T-tau: total tau, P-tau: phosphorylated tau 181, Aβ-42: amyloid β 42, Aβ-40: amyloid β 40, CDR: Clinical Dementia Rating, MoCA: Montreal Cognitive Assessment).

Sample Type	CSF		
**Participants**	**HC (*n* = 48)**	**MCI (*n* = 38)**	***p*-Value**
**Age (mean ± SD)**	68.92 ± 6.14	75 ± 9.51	0.0003
**Sex** **(% female)**	54.16%	55.30%	Not significant
**Years of Education (mean ± SD)**	16.2 ± 2.29	16.31 ± 2.75	Not significant
**Ethnicity**	Caucasian: 85.4%, African American: 14.6%, Asian: 0%	Caucasian: 95%, African American: 5%, Asian: 0%	Not significant
**Sum of Boxes CDR (mean ± SD)**	0.05 ± 0.25	2.80 ± 1.35	<0.0001
**Global CDR (mean ± SD)**	0.0 ± 0.0	0.5 ± 0.0	<0.0001
**MoCA (mean ± SD)**	26.48 ± 2.60	21.49 ± 4.86	<0.0001
**p-tau181 (mean ± SD)**	33.31 ± 9.48	85.8 ± 47.3	<0.0001
**T-tau (mean ± SD)**	267.0 ± 197	587.4 ± 312.3	<0.0001
**Aβ_1–42_ (mean ± SD)**	1169.7 ± 384.7	723.6 ± 413.0	<0.0001
**Aβ_1–40_ (mean ± SD)**	12,190 ± 3356.6	12,657.0 ± 4978.9	Not significant
**Aβ_1–42_/Aβ_1–40_**	0.0956 ± 0.015	0.058 ± 0.020	<0.05
**Apoe4 Genotype**	21%	58%	<0.05

## Data Availability

The data presented in this study are available in the Appendix A.

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
