# Peer review of "Exploratory Assessment of Proteomic Network Changes in Cerebrospinal Fluid of Mild Cognitive Impairment Patients: A Pilot Study"

_biomolecules, 2023, doi:10.3390/biom13071094_

Round 1

Reviewer 1 Report

Kamalian and coauthors have in their manuscript described a pilot proteomics analysis of cerbrospinal fluid from patients from prestages of Alzheimer's disease, ie mild cognitive impariment compared with cerebrospinal fluid from control persons. The authors have written a straightforward manuscript and the data is presented clearly, although figures need enlarged fonts and higher resolution diagrams. The study is interesting as it investigated differences in protein expressionbetwen control and an AD prestage before manifest disease is present. 

Unfortunately the zip-file with supplementary tables was found empty/invalid when I had downloaded it so this rewiew is unfortunately only taking the images in the main manusript text into account. 

Feedback per section:

Introduction: This section is concise and clear.

Material and Methods:

Lines 78-80: Please state the cutoff values used for amyloid beta 42/40 ratios as well as p-tau. 

Results:

Table 1: Was there any significant statistical differences in ethnicity beween the two groups?

Please check that all the abbreviations in the table are explained in the table legend (such as CDR, MoCA, 

Figure 1: Please provide a higher-resolution image of this figure. Also, "normal" should say HC above the y-axis (HC is explained in the  . Also enlarge all text in this figure, such as table axes and legend explanations (to the sizes of Figure 2 which reads better). 

The significance of triangle symbols instead of dots have not been explained in the legend. I assume that they are used to point at gene names/data points although it is not clear how some proteins have ben selected to be pointed out among the red points and some are not named. 

Lines 173-174, 177-178, 193, 196-197, 199-200, 203-204, 206 and 216: Please explain abbreviations of these proteins. 

Lines 180-182: It is unclear if the multivatiate linear regression was done on the whole dataset or only on specific proteins. Please clarify. 

Figure 3, panels a and c need to be enlarged to allow for reading the text. Also, the significance of grey vs yellow protein names is not clear in figure 3C. In figure 3c, it is also unclear how the vertical lined proteins were selected to be presented.  

Discussion:

Line 223-224, 246-250 and onwards in the Discussion: the statistics fits better to be presented in the Results section, which is at this point pretty short and presentation of these individual statistics of the proteins could be moved there. Now the reading of the discussion is a bit imparied by the results/statistics mentioned here. 

Line 233: Specify which supplemental table or figure the text is refering to. 

Author Response

Dear reviewer,

Please find below a detailed response to each of the brought up comments.

Feedback per section:

Introduction: This section is concise and clear.

Material and Methods:

Lines 78-80: Please state the cutoff values used for amyloid beta 42/40 ratios as well as p-tau. 

-Thank you for your thoughtful comment. We added the cutoffs used to choose the participants.

Results:

Table 1: Was there any significant statistical differences in ethnicity beween the two groups?

-Thank you for your attention to detail. No, there weren’t significant (p-value = 0.15), and we included it in the table.

Please check that all the abbreviations in the table are explained in the table legend (such as CDR, MoCA, 

-Thanks for your reminder. We wrote the abbreviations in the table legend.

Figure 1: Please provide a higher-resolution image of this figure. Also, "normal" should say HC above the y-axis (HC is explained in the  . Also enlarge all text in this figure, such as table axes and legend explanations (to the sizes of Figure 2 which reads better). 

-Thank you for pointing this out. We uploaded a higher resolution image of this figure and replaced the texts as requested in Figure 1.

The significance of triangle symbols instead of dots have not been explained in the legend. I assume that they are used to point at gene names/data points although it is not clear how some proteins have ben selected to be pointed out among the red points and some are not named. 

-Thank you for your comment. We have clarified that the triangles are connectors that point to the label of each protein. The limiting factor to the labels of this figure is space since including the names of all significant proteins would not be feasible. So, the algorithm will refrain from including the labels that overlap. The detailed list of significantly up- and downregulated proteins and their log fold changes is listed in the supplementary table s1 in detail.

Lines 173-174, 177-178, 193, 196-197, 199-200, 203-204, 206 and 216: Please explain abbreviations of these proteins. 

-Thank you for your comment. We have included the full protein name of all the proteins assayed by Olink as reported by Olink in a supplementary table s5 for the reader’s reference.

Lines 180-182: It is unclear if the multivatiate linear regression was done on the whole dataset or only on specific proteins. Please clarify. 

-The multivariate linear regression was performed on the whole dataset and the results for the whole dataset are included in the supplementary table.

Figure 3, panels a and c need to be enlarged to allow for reading the text. Also, the significance of grey vs yellow protein names is not clear in figure 3C. In figure 3c, it is also unclear how the vertical lined proteins were selected to be presented.  

-Thank you for your suggestions. We have replaced Figure 3 according to your suggestions. The vertical line proteins are chosen based on the protein pathway dataset (The granulocyte pathway in the BioCarta functional database). The enrichment of these proteins known to belong to granulocyte pathway based on this database point to enrichment of this pathway in MCI patients. The horizontal lines of the 3C figure are the subject IDs of participants belonging to MCI (grey) and healthy controls (yellow). We have clarified this in the figure based on your comment.

Discussion:

Line 223-224, 246-250 and onwards in the Discussion: the statistics fits better to be presented in the Results section, which is at this point pretty short and presentation of these individual statistics of the proteins could be moved there. Now the reading of the discussion is a bit imparied by the results/statistics mentioned here. 

-Thank you for your suggestion. We have deleted the p-values and log fold changes of the proteins in the discussion section.

Line 233: Specify which supplemental table or figure the text is refering to. 

-Thanks for pointing this out. The power analysis is provided in the Methods section of the Supplementary Material file page 2.

Reviewer 2 Report

Re biomolecules-2470947

This is an elegant work studying various proteins in the CSF of early AD. The authors observed that secernin-1 (a tau binding protein), microtubule-associated proteins, MAPK signaling-related proteins, protein missfolding-related molecules and neuroinflammation-related proteins are implicated in mechanisms/pathways related to AD pathogenesis in the early stages. Such studies are important because they help improving our understanding of AD pathogenesis, but additionally, they may prove helpful in identifying biomarkers for the correct diagnosis of early AD. The study is well conducted and statistics are excellent.

One minor point, Page 2 line 77: Please provide here the cut-offs of your lab for Aβ42/40 ratio and p-tau181.

Author Response

Dear reviewer,

Please find attached our response to the brought up comments.

This is an elegant work studying various proteins in the CSF of early AD. The authors observed that secernin-1 (a tau binding protein), microtubule-associated proteins, MAPK signaling-related proteins, protein missfolding-related molecules and neuroinflammation-related proteins are implicated in mechanisms/pathways related to AD pathogenesis in the early stages. Such studies are important because they help improving our understanding of AD pathogenesis, but additionally, they may prove helpful in identifying biomarkers for the correct diagnosis of early AD. The study is well conducted and statistics are excellent.

One minor point, Page 2 line 77: Please provide here the cut-offs of your lab for Aβ42/40 ratio and p-tau181.

-Thank you for your thoughtful comment. We have clarified the cutoffs of these biomarkers in the methods section.